# Self-reported symptom burden among patients attending public health care facilities in India: Looking through ICPC-3 lens

**Priti Gupta**[1], **Bhavna Bharati**[2], **Kirti Sundar Sahu**[2], **Pranab Mahapatra**[3], **Sanghamitra Pati**[4]*

1 Research Department, Centre for Chronic Disease Control, New Delhi, India, 2 Department of Public Health, Bhubaneswar Advanced Rehabilitation Center, Bhubaneswar, Odisha, India, 3 Department of Psychiatry, Kalinga Institute of Medical Sciences, KIIT University, Bhubaneswar, Odisha, India, 4 Department of Health Research, ICMR-Regional Medical Research Centre, Bhubaneswar, Odisha, India

* drsanghamitra12@gmail.com

**Data Availability Statement:** Deidentified data is uploaded as supporting file.

**Funding:** This study was financially supported by funding from the Department of Health & Family Welfare, Goverment of Odisha, India in the form of

## Abstract

### Introduction

The objectives of this study were: 1) to describe the socio-demographics and classify the chief complaints and reasons to encounter facilities of patients presenting to public health-care facilities; 2) to explore differences in these complaints and: International Classification of Primary Care-3 (ICPC-3) groups across socio-demographic and health system levels.

### Methods

This is a cross-sectional study conducted in three districts of Odisha, India. Within each district, the district hospital (DH), one Sub-district hospital (SDH) (if available), two Community health centers (CHCs), and two Primary health care centers (PHCs) were selected. Thus, a total of three DHs, three SDHs, six CHCs, and six PHCs were covered. Two tertiary health-care facilities were also included. Patients aged 18 years and older, attending the Outpatient Departments (OPD) of sampled health facilities were chosen as study participants through systematic random sampling.

### Results

A total of 3044 patients were interviewed. In general, 65% of the sample reported symptoms as their chief complaint for reason of encounter, whereas 35% reported disease and diagnosis. The most common reasons to encounter health facilities were fever, hypertension, abdominal pain, chest pain, arthritis, skin disease, cough, diabetes, and injury. Among the symptoms, the highest number of patients reported the general category (29%), followed by the digestive system (16%). In the disease category, the circulatory system has the highest proportion, followed by the musculatory system. In symptom categories, general, digestive, and musculatory systems were the key systems for the reasons of encounter in outpatient departments irrespective of different groups of the population. In terms of different tiers of

a grant (308/SHRMU) for data collection and analysis received by SP. No additional external funding was received for this study. The funder had no role in study design, data collection and analysis, decision to publish, or preparation of the manuscript.

**Competing interests:** The authors have declared that no competing interests exist.

health systems, the top three reasons to visit OPD were dominated by the circulatory system, respiratory system, and musculatory system.

## Conclusion

This is the first Indian study using the ICPC-3 classification for all three levels of health care. Irrespective of age, socio-economic variables, and tiers of healthcare, the top three groups to visit public health facilities according to the ICPC-3 classification were consistent i.e., general, digestive, and circulatory. Implementation of standard management and referral guidelines for common diseases under these groups will improve the quality and burden at public health facilities in India.

## Introduction

### India public health system

India's public health care system has a three-tier structure comprising primary, secondary, and tertiary levels. Primary health care services are provided by sub-centers and primary health centers (PHCs), while community health centers (CHCs) and sub-divisional hospitals (SDH) render secondary care. Tertiary health care is provided by district hospitals (DH) and medical college hospitals under the directorate of medical education [1]. While secondary and tertiary levels of care are usually shared between rural and urban populations, primary level care is different between rural and urban India. Unlike rural areas, there are no sub-centers in urban areas and community outreach services are provided by female health workers, who report to urban PHC [2].

### Why symptom burden is important?

Symptom is a subjective term where the patient experiences the disease or physical disturbances rather than the doctor noticing them [3]. Usually, patients report symptoms at the first visit which are later investigated to diagnose a disease. However, there are studies suggesting that many physical symptoms remain unexplained even after thorough evaluation [4,5] Inability to address the symptoms has a significant impact on the quality of life, which leads to both psychological stress and functional decline among patients [6]. Therefore, symptom reporting and classification in a standard way is one of the important criteria for strengthening primary care.

### Why to use ICPC-3 classification?

Keeping in mind the different needs of physicians and covering the gaps in the International Classification of Diseases (ICD), the International Classification of Primary Care (ICPC) was introduced mainly to understand the reason for contact with primary care [7]. The International Classification of Primary Care (ICPC) is the most widely used international classification for systematically capturing and ordering clinical information in primary care. It was reviewed and launched in December 2020 by the ICPC-3 Consortium in December 2020 [8]. ICPC is the most effective tool by which the needs of the patient can be assessed by gathering all the information and correlating it. In primary care, the physician comes in contact with a patient multiple times until diagnosis. ICPC helps in capturing the episodes of care (EoC). The collective gathering of information can help in the quality of care, speedy diagnosis, and

increase the knowledge of primary care practitioners [8]. For example, a single person has come in contact with the physician three times for a diagnosis, however, if there is a proper ICPC then the contact time can be minimized and there will be speedy diagnosis and treatment. Proper ICPC use not only saves physician time but also the number of patient burdens in an outpatient. This also increases the efficiency and quality of the physician in the Primary health care. There was always a need for an international standard of reporting and monitoring disease which can lead to a common platform for and allows the physicians to share data in a standard way [9]. As always, the cause of mortality and morbidity is changing in that case an effective implementation of ICPC will be an effective tool in many ways. If we monitor and document the symptom burden of any country, then there can be equitable planning in health care distribution by giving the area what they need.

The latest version of the International Classification of Primary Care (ICPC-3), published in 2021 [10] represents a substantial improvement over the previous edition [11]. Unlike its predecessors, ICPC-3 caters to all primary care clinicians, not just family physicians or general practitioners. Its interoperability with various classifications, such as ICD-10, ICD-11, ICHI, ICF, and SNOMED CT, allows seamless integration and information exchange. Embracing ICPC-3 offers a logical and comprehensive approach to organizing and conceptualizing primary care within a unified classification system [12].

The objectives of this study were: 1) to describe the socio-demographics of patients presenting to public healthcare facilities; 2) to document chief complaints and reasons to visit these facilities and classify them using ICPC-3; and 3) to explore differences in these complaints and ICPC-3 groups across socio-demographic and health system levels.

## Methods

This cross-sectional study was conducted in Odisha, India. According to the revenue zones of the state, the state is divided into three divisions, such as the central division (C.D.), the north division (N.D.), and the south division (S.D.) with 10 districts in each division. There is one DH, 1 or 2 SDH, and 3–6 CHCs in each district. It was planned to select three districts, comprising one from each revenue division. Selected districts of Odisha were Cuttack, Sambalpur, and Nabarangapur, with proportionate representation from each type of facility. The only exception was Nabarangapur district, as there was no medical college. Within each district, the district hospital, one SDH (if available), two CHCs, and two PHCs were selected. Thus, a total of three DHHs, three SDHs, six CHCs, and six PHCs were covered. Care was taken to include at least one non-communicable disease (NCD) program implemented in each district. Two tertiary healthcare facilities, Veer Surendra Sai Institute of Medical Sciences and Research, Burla formerly known as (VSS MCH), and Srirama Chandra Bhanja Medical College and Hospital, Cuttack (SCB MCH) were also included. A total of 3377 patients were proposed to be interviewed throughout the study, with proportionate representation from each facility. It was decided to include nearly 160 patients from PHC, 240 patients from CHC, 330 patients from DHH, and 850 patients from MCH, thus making a total of 3040 from three districts. After considering a non-response rate of 10%, 3377 patients were proposed to be interviewed over a three-month data collection period. Adult patients (more than 18 years) attending the Outpatient Departments (OPD) of sampled health facilities were chosen as study participants through systematic random sampling. A total number of eligible patients was recorded during one week of consultation at each site; the proportion of attendees in one week can be calculated in our sample accordingly. Informed consent from the participant was obtained before the interview. Participants were interviewed using a predesigned and pretested (both Odia and English versions) questionnaire while they were waiting to see their doctor or immediately

after the consultation. The Odia version of the instrument was pretested for validity on a small sample of non-study patients, and necessary revisions were made. The instrument was then backtranslated into English, and the questionnaire was evaluated for fidelity to the original intent or purpose of asking each question. Two trained field investigators collected the data. The data was collected over three months in 2016.

Certain categories of patients were excluded from self-reporting, such as those too ill to participate, those with insufficient cognitive ability, or those with any physical or mental disabilities; hence data was collected by interviewing their attendants. To avoid duplication, any patient who had already been interviewed in any of our selected facilities was excluded. The utmost care was taken to maintain the confidentiality of the data shared by the patient.

An extensive review of the available literature was undertaken to gather all research studies on morbidity, its estimation, measurement, and correlates, and extract relevant information. Input from the expert group were incorporated. The ICPC-3 coding of diseases, symptoms, and reasons for the visit was followed by two experienced researchers and validated with the senior researcher for any discrepancies.

The study tool includes socio-demographic information (age, sex, residence, ethnicity, religion, educational level (number of years of schooling), marital status, monthly income from all sources, monthly household expenditure, current housing, and household composition, and insurance status including, Rashtriya Swasthya Bima Yojana (RSBY) and other insurances.

Unique chief complaints were grouped and coded under ICPC-3. Only the chief complaint results are presented in this paper. For some of the common symptoms, probable ICPC-3 codes were allocated as differential diagnoses for critical analysis. Once consented to by the authors, symptom codes were finalized, and new variables were created for further analysis.

After cleaning the data, descriptive analysis was done to assess differences in symptom presentation across demographic indicators (age, sex, education, place of living), three levels of health care, behavioral risk factors (smoking, alcohol), and the presence of chronic diseases. Data visualization tools were used to explore the clustering of symptoms by the system. Data entry was completed in SPSS, and data cleaning and analysis were done using R software.

Ethical approval to conduct this study was obtained from the Public Health Foundation of India (PHFI) research ethics committee (approval number is 288/State Research and Ethics Committee/13/11/2014). The necessary prior official permission was obtained to conduct the study at the public health facility. The purpose and procedure of the study were disclosed to the participants and informed consent was obtained from them. All steps were taken to ensure that confidentiality and anonymity were maintained at all times.

## Results

In total, 3044 patients were interviewed. The ICPC-3 classification has been conducted for the samples with valid chief complaints only, so further analysis has been done for the sample of 2568, as 476 respondents have not reported any chief complaints. As the chief complaints of three patients were unable to comprehend, the data analysis excluded those subjects, and the final data was analyzed for 2565 patients.

We have combined samples from PHC and CHC and named them as primary, DHH as secondary, and MCH as tertiary.

The sociodemographic characteristics of the study participants shown in Table 1 illustrate that 50% of the sample were in the 18–39 years age group. Around 53% of the sample were male, 58% were from the general category, and 69% resided in rural areas. More than a quarter did not have any formal education (29%). More than half belong to below poverty line

**Table 1. Demographic features of the participant.**

| Characteristics | Groups | Total N (%) | Primary N (%) | Secondary N (%) | Tertiary N (%) |
|---|---|---|---|---|---|
| Age Group (in years) | 18-39 | 1324(51.6) | 361(51.5) | 679(51.8) | 284(51.3) |
| | 40-59 | 860(33.5) | 221(31.5) | 419(32) | 220(39.7) |
| | 60+ | 381(14.9) | 119(17) | 212(16.2) | 50(9) |
| Gender | Female | 1214(47.3) | 373(53.2) | 552(42.1) | 289(52.2) |
| | Male | 1351(52.7) | 328(46.8) | 758(57.9) | 265(47.8) |
| Caste | SC | 514(20) | 145(20.70) | 266(20.3) | 103(18.6) |
| | ST | 577(22.5) | 251(35.8) | 264(20.2) | 62(11.2) |
| | General | 1474(57.5) | 305(43.5) | 780(59.5) | 389(70.2) |
| Residence | Urban | 912(35.6) | 56(8) | 618(47.2) | 238(43) |
| | Rural | 1653(64.4) | 645(92) | 692(52.8) | 316(57) |
| Education | No formal education | 758(29.6) | 217(31) | 426(32.5) | 115(20.8) |
| | Primary | 955(37.2) | 252(35.9) | 453(34.6) | 250(45.1) |
| | Secondary | 662(25.8) | 193(27.5) | 316(24.1) | 153(27.6) |
| | Graduation and above | 190(7.4) | 39(5.6) | 115(8.8) | 36(6.5) |
| Socioeconomic status | APL | 1260(49.1) | 266(37.9) | 668(51) | 326(58.8) |
| | BPL | 1305(50.9) | 435(62.1) | 642(49) | 228(41.2) |
| Social Insurance | No | 1542(60.1) | 426(60.8) | 757(57.8) | 359(64.8) |
| | Yes | 915(35.7) | 244(34.8) | 491(37.5) | 180(32.5) |
| | Did not know | 108(4.2) | 31(4.4) | 62(4.7) | 15(2.7) |

SC- Schedule Caste, ST- Schedule Tribe, APL – Above Poverty Line, BPL- Below Poverty Line.

category (53.2%) and approximately two-thirds of the sample had no health insurance coverage for health expenditures (60%).

ICPC-3 divides reasons for a visit to the primary health centers into 19 chapters, out of which our sample reported 15 categories (Table 2). Chapters like social problems (ZC), intervention and processes, and functioning related (2F) have not been reported in outpatients in our sample. Each chapter is divided into two components. The components deal with (S) symptoms and complaints; and (D) diseases and diagnosis. In general, 65% of the sample reported symptoms as their chief complaint for reason of encounter, whereas 35% reported disease and diagnosis. Among different categories, the highest number of patients reported for the general category (29%) followed by the digestive system (16%) which is reflected in the symptom category. In the disease category, the circulatory system has the highest proportion, followed by the musculatory system.

The reasons for visits with different demographic and related variables have been described in Fig 1. In the 18–39 age group, pregnancy and childbearing are the most frequent causes of visits to the facility, whereas in both the 40–59 as well 60+ age groups, the circulatory and musculatory systems were the main causes. Both males and females present highest in the circulatory system. In the second position, females have pregnancy and childbearing-related visits, and males have musculatory system-related symptoms. Across different categories of caste, the general category is coming for mainly circulatory, musculatory, and respiratory problems whereas, in the SC & ST categories, the primary reason is a general complaint. In Urban residents, along with the circulatory and musculatory system frequent respiratory problem is common. In rural areas, general problems are more common. No difference was found across different educational categories. In wealthy populations, respiratory complaints predominate over the musculatory and circulatory systems. In symptom categories in general, the digestive

**Table 2. Stratification of reasons for encounter by ICPC– 3 system and subcategories.**

| ICPC-3 Groups | Total N (%) | D N (%) | S N (%) | Primary N (%) | Secondary N (%) | Tertiary N (%) |
|---|---|---|---|---|---|---|
| General | 738(28.8) | 116(4.5) | 622(24.2) | 250(35.7) | 316(24.1) | 172(31) |
| Digestive system | 418(16.3) | 63(2.5) | 355(13.8) | 98(14) | 219(16.7) | 101(18.2) |
| Musculatory system | 393(15.3) | 158(5.9) | 241(9.4) | 96(13.7) | 227(17.3) | 70(12.6) |
| Respiratory system | 214(8.3) | 113(4.4) | 101(3.9) | 71(10.1) | 115(8.8) | 28(5.1) |
| Circulatory system | 190(7.4) | 178(6.9) | 12(0.5) | 46(6.6) | 111(8.5) | 33(6) |
| Skin | 150(5.8) | 38(1.5) | 112(4.4) | 40(5.7) | 76(5.8) | 34(6.1) |
| Neurological system | 140(5.5) | 28(91.1) | 112(4.4) | 25(3.6) | 68(5.2) | 47(8.5) |
| Endocrine system | 87(3.4) | 86(3.4) | 1(0.0) | 9(1.3) | 51(3.9) | 27(4.9) |
| Pregnancy and Childbearing | 80(3.1) | 80(3.1) | 0(0) | 35(5) | 44(3.4) | 1(0.2) |
| Urinary system | 41(1.6) | 14(0.5) | 27(1.1) | 3(0.4) | 23(1.8) | 15(2.7) |
| Eye | 39(1.5) | 13(0.5) | 26(1) | 10(1.4) | 20(1.5) | 9(1.6) |
| Ear | 27(1.1) | 0(0) | 27(1.1) | 10(1.4) | 13(1) | 4(0.7) |
| Blood | 23(0.9) | 23(0.9) | 0(0) | 1(0.1) | 20(1.5) | 2(0.4) |
| Genital system | 22(0.9) | 3(0.1) | 19(0.7) | 6(0.9) | 5(0.4) | 11(2) |
| Psychological system | 3(0.1) | 2(0.1) | 1(0.0) | 1(0.1) | 2(0.2) | 0(0) |
| Total | 2565(100) | 909(35.4) | 1656(64.6) | 701(100) | 1310(100) | 554(100) |

Note: D- Diagnosis and diseases, S- Symptoms, ICPC- International Classification of Primary Care.

and musculatory systems were the key systems for the reason for visits in outpatient departments irrespective of different groups of the population (Table 3).

In terms of different tiers of the health system, the top three reasons to visit OPD were dominated by the circulatory system, respiratory system, and musculatory system. Patients with diabetes are visiting more to the highest level of health facilities which is not reflected in other health facilities. Though a gatekeeping mechanism is in place, still a significant proportion of

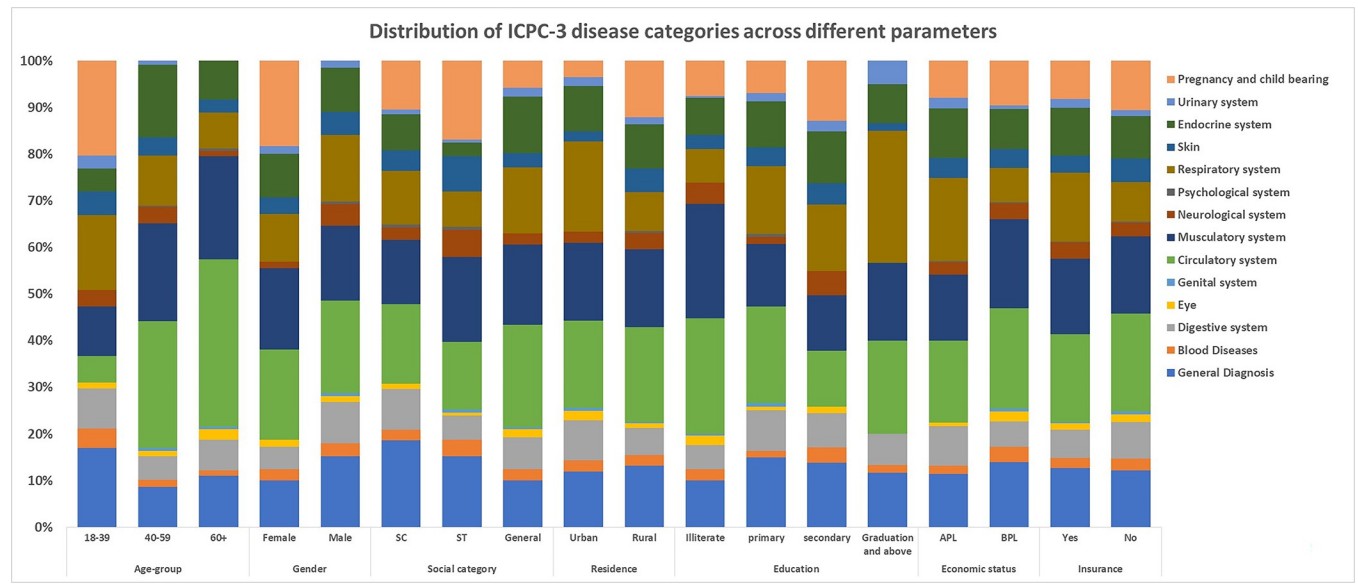

**Fig 1. Distribution of ICPC-3 disease categories across different parameters.**

**Table 3. Top three reasons for visits by demographic features by ICPC-3.**

| | | General N (%) | Digestive N (%) | Muscular N (%) | Total N (%) |
|---|---|---|---|---|---|
| Age Group (in years) | 18-39 | 361(38.8) | 183(19.7) | 115(12.4) | 931(100) |
| | 40-59 | 190(36.2) | 132(25.1) | 88 (16.8) | 525(100) |
| | 60+ | 71 (35.5) | 40 (20) | 38 (19) | 200 (100) |
| Gender | Male | 297(33.8) | 212(24.1) | 146(16.6) | 878 (100) |
| | Female | 325(41.8) | 143(18.4) | 95 (12.2) | 778 (100) |
| Caste | SC | 122(36.7) | 71 (21.4) | 48 (14.5) | 332 (100) |
| | ST | 199(49) | 81 (20) | 31 (7.6) | 406 (100) |
| | General caste | 301(32.8) | 203(22.1) | 162(17.6) | 819 (100) |
| Residence | Urban | 179(32.3) | 117(2.1) | 113(20.4) | 554 (100) |
| | Rural | 443(40.2) | 238(21.6) | 128(11.6) | 1102(100) |
| Education | No formal education | 192 (41) | 100(21.4) | 70 (15) | 468 (100) |
| | Primary | 246(40.1) | 135 (22) | 91 (14.8) | 613 (100) |
| | Secondary | 153(34.4) | 98 (22) | 57 (12.8) | 445 (100) |
| | Graduation and above | 31 (23.8) | 22 (16.9) | 23 (17.7) | 130 (100) |
| Socioeconomic status | APL | 285(34.7) | 173 (21) | 124(15.1) | 822 (100) |
| | BPL | 337(40.4) | 182(21.8) | 117(14.0) | 834 (100) |
| Levels of healthcare | Primary | 221(45.6) | 86 (17.7) | 58 (12) | 485 (100) |
| | Secondary | 253(32.6) | 185(23.8) | 133(17.1) | 777 (100) |
| | Tertiary | 148(37.6) | 84 (21.3) | 50 (12.7) | 394 (100) |

SC- Schedule Caste, ST- Schedule Tribe, APL – Above Poverty Line, BPL- Below Poverty Line.

the population visits medical colleges and hospitals for general conditions such as abdominal pain, vomiting, fever, etc.

Fig 2 explores the variation in the top 16 reasons for visits across different levels of healthcare facilities. Fever is the most common reason across all health facilities. Type 2 diabetes and skin disease are most commonly reported at the district hospital. General abdominal pain is reported at the district and medical college level. Among the top reasons only fever, pregnancy, and hypertension are handled at the primary health care level.

Fig 3: Word cloud explores the reason of most common reasons for a patient to visit a public health facility. The most common reasons are fever, hypertension, abdominal pain, chest pain, arthritis, skin disease, cough, diabetes, and injury.

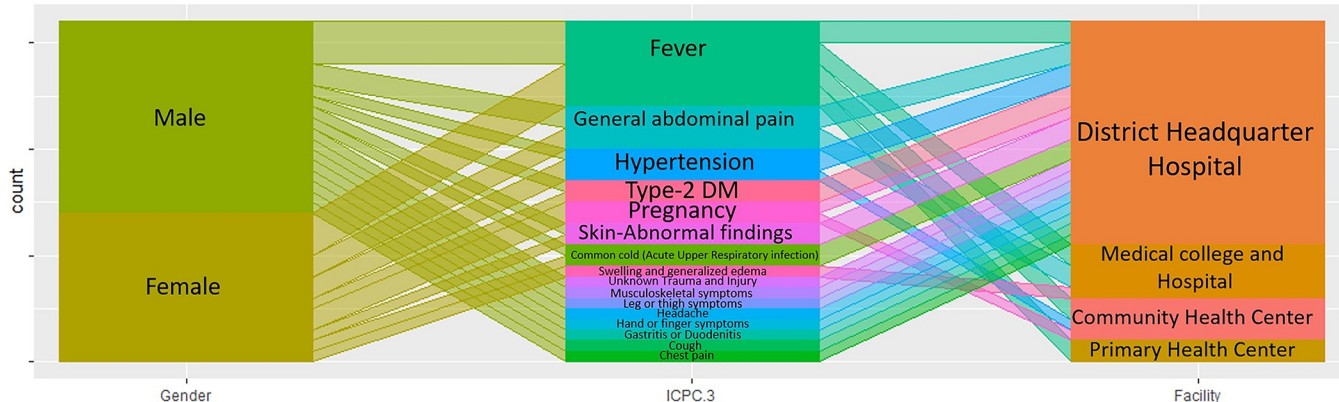

**Fig 2. Top 16 reasons for visits by facilities.**

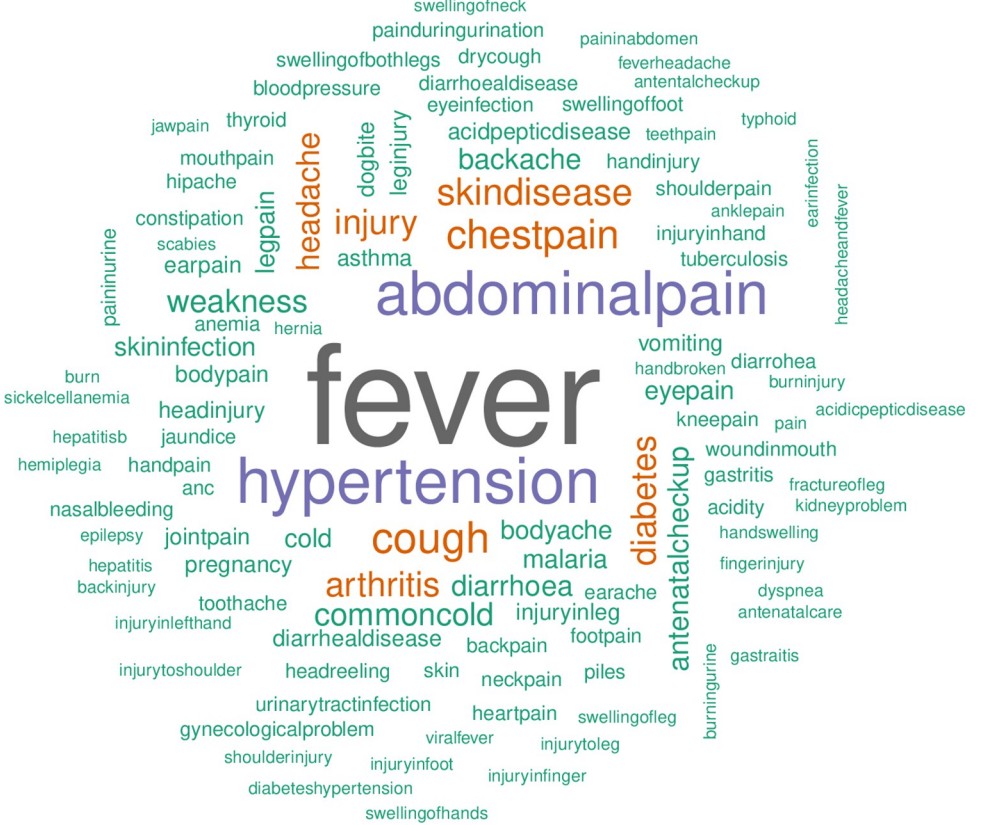

**Fig 3. Word cloud of patient-reported reasons for visits.**

## Discussion

The study indicates that the majority of patients of the public health out-patient services were of younger age group, educated up to primary, and rural residents. The most common complaints in all three tiers are general, muscular, and digestive systems. Symptoms of the circulatory and muscular system increase with age and the respiratory system decreases with age. Interestingly women are to be availing health services at par with their male counterparts irrespective of their education status. Traditionally the healthcare-seeking behavior of the females and expenditure have remained below optimal [13,14]. Findings from this study are indicative of familiarization of healthcare services among women, potential reasons could be increased involvement of frontline workers, women's self-help groups, and various state health policies like Odisha health equity strategy, Odisha livelihood mission, etc. [15,16]. Also, few other studies from India outpatient departments show equal distribution of both gender [17,18]. Like our study majority of patients availing health services in public health facilities were from a productive age group [17–20]. Even though 50% of our study participants were below the poverty line and were supposed to be covered under Rashtriya Swasth Bima Yojana (RSBY) in India [21], about 60% of the population wasn't covered under any health insurance scheme in our study, this finding is similar to other studies from India [22–24]. Though, studies from South India show high health insurance awareness [25,26].

Poor utilization of public primary health services in urban areas could be because of the exclusion of urban PHCs from our study. However the huge difference between rural and urban highlights the need to establish a strong urban primary healthcare in India [27]. The majority of

people utilizing primary care are from BPL families while in secondary and tertiary care there is equal distribution. Other studies from India also show that the majority of patients belonged to lower socioeconomic classes [17,20,28]. As most people visit primary health facilities for pregnancy-related issues, this shows that the number of encounters related to pregnancy and childbearing are part of the primary care services in a programmatic mode. Other studies from India also show that women have more trust and better care at a lower level of health facilities, especially for uncomplicated pregnancies [29]. The National Family and Health Survey (NFHS) also highlights the improved maternal and child care in the state [30]. But, among the other top reasons to visit health facilities, primary health facilities are accessed only for fever and hypertension. This highlights the low level of facilities at the primary health care level in India.

The most common reasons to visit public health facilities are fever, abdominal pain, chest pain, diabetes, hypertension, arthritis, skin disease, cough, and injury. The pattern of dual burden of disease is reflected among our sample as the earliest symptom of any infection is fever which is the foremost reason to visit OPD in all levels of facilities followed by hypertension. Our previous study has also listed these as common diseases [18,31].

There is a high burden of skin diseases in India with very limited facilities for primary health care. Training of allopathic doctors in dermatology during their graduate degree is still minimal, therefore management skills among graduate doctors for these diseases remain subtle [32]. In addition, there is a scarcity of dermatologists in India. As most of the common skin disease diagnoses are based on visual inspection, tele-dermatology could be one way to fill the existing gap in primary care [33].

Irrespective of age, socio-economic variables, and tiers of healthcare top three groups to visit public health facilities according to ICPC-3 classification were consistent i.e., general, digestive, and circulatory. So primary care and health insurance schemes should address these three issues adequately for comprehensive health care. Low reporting of psychological symptoms does not reflect the true burden. This could be owing to two factors, first lack of mental health literacy and appraisal of own or family member's mental symptoms, and second lack of ability of primary care physicians to recognize psychological symptoms as a result psychological symptoms are still associated with stigma within society. All these have resulted in a cumulative disconnect between psychiatry care and primary health services. Our previous study has shown that people prefer to directly consult a mental health specialist for their psychological symptoms rather than approaching primary health care [34]. Thus, circumventing public healthcare systems.

This is the first study from India reporting symptom burden at all three levels of public health facilities in India using the ICPC 3 classification. However, there are a few limitations; firstly, as the study is a part of a government project, the sample size calculation is not based on ideal statistical based sample size calculation but based on the project objectives; secondly, the principle of primary healthcare is holistic which should include physical mental and social. Accordingly, ICPC-3 has kept a chapter on social problems. However, we could not find any reported reason for visits in this chapter in our study participants. System appraisal for cognitive functioning and ADL appears to be low since participants did not mention it in their chief complaint. One of the reasons could be our questionnaire did not have any probes to elicit such complaints. Social problems including domestic violence, neglect, abuse, and self-harm are less likely to be reported to primary care. Also, we have excluded Urban-PHCs in our sample, therefore, there is an underreporting of the urban poor in our study.

## Conclusion

This is the first Indian study using ICPC-3 classification for all three levels of health care. Irrespective of age, socio-economic variables, and tiers of healthcare top three groups to visit

public health facilities according to ICPC-3 classification were consistent i.e., general, digestive, and circulatory. This high symptom burden of the general, digestive, and circulatory system demands proper clinical management and referral guidelines for these diseases at the primary healthcare level for timely management and to decrease the burden of higher health facilities. The high burden of skin diseases and low reporting of mental health problems at primary health care necessitates primary care physician capacity building and incorporation of specialist tele-consultation for these diseases at the primary health care level. ICPC-3 can be used to improve the registration process in hospitals and can be further used for this kind of research, as for every patient defining an ICD coding at the time of registration is difficult.

## Supporting information

**S1 Checklist. STROBE statement—checklist of items that should be included in reports of observational studies.**
(DOCX)

**S1 Questionnaire.**
(PDF)

**S1 Data.**
(XLSX)

## Acknowledgments

The authors are grateful to the state health authority, the medical officers of the participating healthcare institutions, and the outpatients attending these facilities for their kind cooperation and timely support. We also are thankful to Dr Subhashisa Swain for his valuable input.

## Author Contributions

**Conceptualization:** Sanghamitra Pati.

**Data curation:** Kirti Sundar Sahu.

**Formal analysis:** Kirti Sundar Sahu.

**Funding acquisition:** Sanghamitra Pati.

**Methodology:** Bhavna Bharati, Kirti Sundar Sahu.

**Project administration:** Bhavna Bharati, Kirti Sundar Sahu, Sanghamitra Pati.

**Resources:** Sanghamitra Pati.

**Supervision:** Sanghamitra Pati.

**Validation:** Bhavna Bharati, Pranab Mahapatra.

**Visualization:** Priti Gupta, Bhavna Bharati, Pranab Mahapatra.

**Writing – original draft:** Priti Gupta, Bhavna Bharati, Kirti Sundar Sahu.

**Writing – review & editing:** Priti Gupta, Pranab Mahapatra, Sanghamitra Pati.

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
