## [Decision Letter · Decision Letter 0]

13 Jun 2023

PGPH-D-23-00452

Self-reported symptom burden among patients attending public health care facilities in India: Looking through ICPC-3 lens

Dear Dr. Pati,

Thank you for submitting your manuscript to PLOS Global Public Health. After careful consideration, we feel that it has merit but does not fully meet PLOS Global Public Health’s publication criteria as it currently stands. Therefore, we invite you to submit a revised version of the manuscript that addresses the points raised during the review process.

I would like to sincerely apologise for the delay you have incurred with your submission. It has been exceptionally difficult to secure reviewers to evaluate your study. We have now received two completed reviews; the comments are available below. The reviewers have raised significant scientific concerns about the study that need to be addressed in a revision.

Please revise the manuscript to address all the reviewer's comments in a point-by-point response in order to ensure it is meeting the journal's publication criteria. Please note that the revised manuscript will need to undergo further review, we thus cannot at this point anticipate the outcome of the evaluation process.

We look forward to receiving your revised manuscript.

Kind regards,

Miquel Vall-llosera Camps

Staff Editor

Journal Requirements:

b. If any authors received a salary from any of your funders, please state which authors and which funders.

2. Please provide separate figure files in .tif or .eps format only and remove any figures embedded in your manuscript file. Please also ensure all files are under our size limit of 10MB.

4. In the online submission form, you indicated that "The data used for this study is available with the corresponding author and can be shared upon reasonable request and prior approval of the State Health & Family Welfare department". All PLOS journals now require all data underlying the findings described in their manuscript to be freely available to other researchers, either 1. In a public repository, 2. Within the manuscript itself, or 3. Uploaded as supplementary information.

Reviewers' comments:

Reviewer's Responses to Questions

**Comments to the Author**

1. Does this manuscript meet PLOS Global Public Health’s publication criteria? Is the manuscript technically sound, and do the data support the conclusions? The manuscript must describe methodologically and ethically rigorous research with conclusions that are appropriately drawn based on the data presented.

Reviewer #1: Partly

Reviewer #2: Partly

2. Has the statistical analysis been performed appropriately and rigorously?

Reviewer #1: I don't know

Reviewer #2: Yes

3. Have the authors made all data underlying the findings in their manuscript fully available (please refer to the Data Availability Statement at the start of the manuscript PDF file)?

Reviewer #1: No

Reviewer #2: No

4. Is the manuscript presented in an intelligible fashion and written in standard English?

Reviewer #1: Yes

Reviewer #2: Yes

5. Review Comments to the Author

Reviewer #1: Before going into detail on reading the content of your submission, I would like to point out a few things:

- What is missing is the argumentation for why you want to use the ICPC-3 for your study? It is mentioned in the introduction, but it bares no evidence. Why use ICPC-3 and not ICPC-2 or ICD01 or ICD-11 or SNOMED CT? Please make your choice more clear.

- There is the mention in the introduction for the WICC being the developers of the ICPC-3: this is not the case. The ICPC-3 has been developed by the ICPC-3 Consortium. You can read about that on the ICPC-3.info website.

- What is missing is the most basic reference" the ICPC-3 Manual and Classification, which is the book.

- The conclusion is not reflecting the results of the study in terms of answering the research questions.

I think your research question could be: Is the ICPC-3 suited for exploring and comparing across and between different levels of health care? The study should be answering this question.

Reviewer #2: This is one of the first attempt to use International Classification of Primary Care in health care facilities of India. The author need to discuss following for better understanding

a. How sample size was estimated and how proportions were estimated from each facility (Line no. 145- 150)

b. how coding was done and aspects of inter- rater reliability must be discussed

c. Does ICP classification provides any additional advantage over ICD classification in Indian setting

d. As fever is one of predominant symptoms , what seasons / months were chosen for data collection from health facility ? Does this choice of month may impact data collection in form of symptoms especially ?

e. Line no. 177-patients with multiple complaints ... separately ".. where this data is provided ?

f. stratification of diseases and symptoms in Table 2 should be done as per health facility rather as separate column

g. How data collection was done from SDH / DH / Medical college - as they have multiple OPDs .. so which area was targeted for one week data collection by investigators ?

6. PLOS authors have the option to publish the peer review history of their article (what does this mean?). If published, this will include your full peer review and any attached files.

**Do you want your identity to be public for this peer review?** For information about this choice, including consent withdrawal, please see our Privacy Policy.

Reviewer #1: No

Reviewer #2: **Yes: **Neeti Rustagi

---

## [Decision Letter · Decision Letter 1]

20 Nov 2023

PGPH-D-23-00452R1

Self-reported symptom burden among patients attending public health care facilities in India: Looking through ICPC-3 lens

Dear Dr. Pati,

Thank you for submitting your manuscript to PLOS Global Public Health. After careful consideration, we feel that it has merit but does not fully meet PLOS Global Public Health’s publication criteria as it currently stands. Therefore, we invite you to submit a revised version of the manuscript that addresses the points raised during the review process.

Only one previous reviewer re-evaluated the revision. Please see the reviewer's comments below.

We look forward to receiving your revised manuscript.

Kind regards,

Jianhong Zhou

Staff Editor

Journal Requirements:

1. Please update your online Competing Interests statement. If you have no competing interests to declare, please state: “The authors have declared that no competing interests exist.”

2. Please provide separate figure files in .tif or .eps format only and ensure that all files are under our size limit of 10MB. You may leave the figure captions or legends in the manuscript.

For more information about how to convert your figure files please see our guidelines: https://journals.plos.org/sustainabilitytransformation/s/figures

3. Please ensure that you refer to Table 3 in your text as, if accepted, production will need this reference to link the reader to the table.

4. Please add a full list of legends for all your Supporting Information files after the References list.

Additional Editor Comments (if provided):

Reviewers' comments:

Reviewer's Responses to Questions

**Comments to the Author**

1. If the authors have adequately addressed your comments raised in a previous round of review and you feel that this manuscript is now acceptable for publication, you may indicate that here to bypass the “Comments to the Author” section, enter your conflict of interest statement in the “Confidential to Editor” section, and submit your "Accept" recommendation.

Reviewer #1: (No Response)

2. Does this manuscript meet PLOS Global Public Health’s publication criteria? Is the manuscript technically sound, and do the data support the conclusions? The manuscript must describe methodologically and ethically rigorous research with conclusions that are appropriately drawn based on the data presented.

Reviewer #1: Yes

3. Has the statistical analysis been performed appropriately and rigorously?

Reviewer #1: Yes

4. Have the authors made all data underlying the findings in their manuscript fully available (please refer to the Data Availability Statement at the start of the manuscript PDF file)?

Reviewer #1: Yes

5. Is the manuscript presented in an intelligible fashion and written in standard English?

Reviewer #1: No

6. Review Comments to the Author

Reviewer #1: I recommend having the submission reviewed on the use of proper English and punctuation which will make it more readable. Some lines have been mentioned below.

It is clearly reflecting the complexity of the Health Care System in India and how ICPC-3 is supporting the purpose it has been designed for: the importance of primary level of care and the exchange of data between tiers from a Primary Health Care perspective.

However, this is not yet reflected in the study objectives in the introduction and is also missing in the conclusion. It pops up somewhere in line 133.

Line 46, 47: add more keywords referring to the content of the article

Line 98: remain

Line 99: has

Line 102: primary care, not the primary care

Line 105: need or needs?

Please check the English in this article on correctness

Line 110: it is not updated but reviewed and launched in December 2020

Line 117, 118: The sentence starting with It not only saves …..what does it mean?

Line 133: The study examining the suitability of ICPC-3 for cross-level exploration is not mentioned in the study objectives in the introduction. It should also be part of the conclusion, stating if the ICPC-3 has shown to be suitable, and why.

Line 155: was proposed = were proposed or were interviewed…again check correctness of English

Line 165: fidelity is not the correct word, should be validity?

Line 170: had already

Line 185,186: not very understandable sentence

Line 222 and 234 : reasons for encounter

Line 224: have not been reported?

Again please check for English

Line 237, 238, 239: Within sex both male and female present highest in

circulatory system whereas females in addition have pregnancy and childbearing related

visits which in male is musculatory reasons in the second position.

Should this read as:

Within sex, both male and female present highest in

circulatory system whereas females in addition have pregnancy and childbearing related

visits, and for male is musculatory reasons in the second position?

A lot of punctuation is missing in all of the article.

Line 275: majority of

Line 281 and 282: Punctuation

Line 303: But not to start a sentence.

Line 321: Starting a sentence with So…….maybe use: We recommend ….

Line 328: has shown or studies have shown

Line 338: reasons for encounter

I do have some additional suggestions regarding the manuscript. PGPH-D-23-00452R1

Self-reported symptom burden among patients attending public health care facilities in India: Looking through ICPC-3 lens

I think it needs to be more explained that the research is not about assessing the useability of ICPC-3, but that the starting point for the exercise was that the ICPC-3 is presumed usable by definition.

In the limitations part of the manuscript it can be addressed as limitations and positive results. There it can be explained that although it had been adopted upfront that the ICPC-3 could be used, in practice it has also been shown that this presumption can be confirmed.

What I think afterwards is that the questionnaire is not presented, which makes it difficult to judge the ICPC-3 principles.

7. PLOS authors have the option to publish the peer review history of their article (what does this mean?). If published, this will include your full peer review and any attached files.

**Do you want your identity to be public for this peer review?** For information about this choice, including consent withdrawal, please see our Privacy Policy.

Reviewer #1: No

---

## [Decision Letter · Decision Letter 2]

22 Jan 2024

PGPH-D-23-00452R2

Self-reported symptom burden among patients attending public health care facilities in India: Looking through ICPC-3 lens

Dear Dr. Pati,

Thank you for submitting your manuscript to PLOS Global Public Health. After careful consideration, we feel that it has merit but does not fully meet PLOS Global Public Health’s publication criteria as it currently stands. Therefore, we invite you to submit a revised version of the manuscript that addresses the points raised during the review process.

We look forward to receiving your revised manuscript.

Kind regards,

Ziyue Wang, M.B.B.S. MPH

Guest Editor

Journal Requirements:

Additional Editor Comments (if provided):

In addition to all the comments from our reviewers, I have three comments that I would like the authors to address before we publish:

1. Each figure should be at sufficient resolution (i.e., 300 dpi at 5 in.) and should be clear sharp images.

2. In the discussion, focus on the strength and added value of this study (e.g. compared to other studies of symptom burden in India or studies using the ICPC-3 in other areas, how does your study add to our collective understanding of symptom burden, especially for our wider readership around the world?).

3. For the limitation, either show some strategies that the authors tried to reduce the impact of the conclusion, or argue why the conclusion is still robust with these limitations.

Reviewers' comments:

Reviewer's Responses to Questions

**Comments to the Author**

1. If the authors have adequately addressed your comments raised in a previous round of review and you feel that this manuscript is now acceptable for publication, you may indicate that here to bypass the “Comments to the Author” section, enter your conflict of interest statement in the “Confidential to Editor” section, and submit your "Accept" recommendation.

Reviewer #1: (No Response)

Reviewer #3: (No Response)

2. Does this manuscript meet PLOS Global Public Health’s publication criteria? Is the manuscript technically sound, and do the data support the conclusions? The manuscript must describe methodologically and ethically rigorous research with conclusions that are appropriately drawn based on the data presented.

Reviewer #1: Yes

Reviewer #3: Yes

3. Has the statistical analysis been performed appropriately and rigorously?

Reviewer #1: Yes

Reviewer #3: Yes

4. Have the authors made all data underlying the findings in their manuscript fully available (please refer to the Data Availability Statement at the start of the manuscript PDF file)?

Reviewer #1: No

Reviewer #3: No

5. Is the manuscript presented in an intelligible fashion and written in standard English?

Reviewer #1: Yes

Reviewer #3: Yes

6. Review Comments to the Author

Reviewer #1: Although the manuscript has improved a lot, the abstract still needs a slight improvement of English.

As for the use of "reason for visits": this needs to be Reason for Encounter to be in line with the principles of the ICPC-3. I think an overactive translator has changed reason for encounter into reason for visits. Please correct this.

What is still missing is the experience of using ICPC-3 for the research. Is ICPC-3 an improvement for registrations or not? That needs to be made clear. Although it is not the research question it can be mentioned in the conclusion that the ICPC-3 is indicated for this kind of research.

I also still miss the questionnaire that has been used. Without it it is hard to see what the results are based on.

Reviewer #3: The article titled "Self-reported symptom burden among patients attending public health care facilities in India: Looking through ICPC-3 lens" investigates the symptom burden in patients using public health facilities in India. The study aims to understand the types of symptoms prevalent in this population and to identify patterns that could inform healthcare delivery improvements in these settings. Despite the significant topic as well as interesting findings, there are some important issues that the authors should address before publication.

Introduction:

1. It is good to know India's public health system in this section. However, with no recent data (either resource allocation, expenditure, or utilization), it remains blank for readers to understand how different levels of the public health system work together.

2. The concept of "symptom burden" was not clearly defined in the article. Is "symptom burden" equal to "symptom reporting and classification"? How do they differ or relate to each other?

3. The objectives of the study do not match in the abstract vs. in the introduction. And the first objective "to describe the socio-demographics of patients presenting to public healthcare facilities" is not necessarily related according to the content.

Methods:

4. The sampling process is detailed and clear. To make it easier for readers, it is recommended that the authors specify the strategy at the beginning (for example, two-stage stratified sampling, convenience sampling, etc) instead of in the middle of a long paragraph. In addition, the inclusion and exclusion criteria should be specified and listed in numbering.

5. Missing data: About 16% of the data were not analyzed due to not reporting any chief complaints. It might be a concern that those observations were not missing at random, and the authors should further specify the randomness (or not) of missing data in the supplement. For example, whether the socio-demographic information differs between having or not having any chief complaints. (See: Marino, M., Lucas, J., Latour, E., & Heintzman, J. D. (2021). Missing data in primary care research: importance, implications and approaches. Family practice, 38(2), 200–203. https://doi.org/10.1093/fampra/cmaa134)

6. Table 1, 2&3: It is recommended to add the outcome in a column for chi-square test or one-way ANOVA, so that the readers can clearly see whether the respondents' characteristics vary across types of public health centers.

Results and discussion:

7. There are many points in the results and discussion sections like gender equity or urban-rural gap issue. However, the key takeaways are not sufficiently and clearly highlighted, especially in the first paragraph of discussion section as well as in the abstract. It is suggested that the authors rephrase some of the paragraphs to emphasize the key points. Accordingly, the analysis in Results could be deepened, such as examining how symptoms differ between men and women or between urban and rural populations.

8. The sample were selected from different districts, and it would be interesting to see whether there are systematic variations across districts using intraclass correlation coefficient (ICC) or other statistic measures.

9. Self-reported symptoms can be subject to various biases, where patients may not accurately remember or report their symptoms. This reliance on patients' perception could impact the accuracy and reliability of the symptom data collected, which should be explained in the limitations.

Minors/format issues:

1. Line 52, 55, etc: In the abstract, the full name for abbreviations should be shown when appearing for the first time

2. Line 148, 340: Unexplained abbreviation (NCD, ADL)

3. Line 343: Duplicated "Secondly"

4. There are many typos and formatting mistakes, please carefully check the manuscript.

7. PLOS authors have the option to publish the peer review history of their article (what does this mean?). If published, this will include your full peer review and any attached files.

**Do you want your identity to be public for this peer review?** For information about this choice, including consent withdrawal, please see our Privacy Policy.

Reviewer #1: **Yes: **Huib Ten Napel

Reviewer #3: No

---

## [Decision Letter · Decision Letter 3]

4 Apr 2024

Self-reported symptom burden among patients attending public health care facilities in India: Looking through ICPC-3 lens

PGPH-D-23-00452R3

Dear Dr Pati,

We are pleased to inform you that your manuscript 'Self-reported symptom burden among patients attending public health care facilities in India: Looking through ICPC-3 lens' has been provisionally accepted for publication in PLOS Global Public Health.

Best regards,

Ziyue Wang, PhD

Guest Editor

Please address this issue before we publish the manuscript: Each figure should be at sufficient resolution (i.e., 300 dpi at 5 in.) and should be clear sharp images.

Reviewer Comments (if any, and for reference):

Reviewer's Responses to Questions

**Comments to the Author**

1. If the authors have adequately addressed your comments raised in a previous round of review and you feel that this manuscript is now acceptable for publication, you may indicate that here to bypass the “Comments to the Author” section, enter your conflict of interest statement in the “Confidential to Editor” section, and submit your "Accept" recommendation.

Reviewer #1: All comments have been addressed

2. Does this manuscript meet PLOS Global Public Health’s publication criteria? Is the manuscript technically sound, and do the data support the conclusions? The manuscript must describe methodologically and ethically rigorous research with conclusions that are appropriately drawn based on the data presented.

Reviewer #1: Yes

3. Has the statistical analysis been performed appropriately and rigorously?

Reviewer #1: Yes

4. Have the authors made all data underlying the findings in their manuscript fully available (please refer to the Data Availability Statement at the start of the manuscript PDF file)?

Reviewer #1: Yes

5. Is the manuscript presented in an intelligible fashion and written in standard English?

Reviewer #1: Yes

6. Review Comments to the Author

Reviewer #1: One thing is still to be corrected: reason of encounter = Reason for Encounter (RfE)

If you use it the first time in the text as Reason for Encounter (RfE), at all the other places you can use RfE.

7. PLOS authors have the option to publish the peer review history of their article (what does this mean?). If published, this will include your full peer review and any attached files.

**Do you want your identity to be public for this peer review?** For information about this choice, including consent withdrawal, please see our Privacy Policy.

Reviewer #1: **Yes: **Huib Ten Napel
